# Effect of Gallium Content and Heat Treatment on the Microstructure and Corrosion Rate of Magnesium Binary Alloys

**Anabel A. Hernández-Cortés** *[ID], **José C. Escobedo-Bocardo**[ID], **Dora A. Cortés-Hernández**[ID] and **José M. Almanza-Robles**

Centro de Investigación y Estudios Avanzados de Instituto Politécnico Nacional Unidad-Saltillo, Av. Industrial Metalúrgica No.1062, Parque Industrial Saltillo-Ramos Arizpe, Ramos Arizpe, Coahuila, C.P. 25900, Mexico
* Correspondence: azucena.hernandez@cinvestav.mx or anabelh772@gmail.com. Tel.: +52-844-4389600

**Abstract:** The microstructure and corrosion rate of as-cast and heat-treated binary Mg-Ga alloys with gallium content ranging from 0.375 to 1.5 wt. % were investigated. The corrosion rate was determined by the weight loss method using a simulated body fluid (SBF). The microstructure of the as-cast alloys showed an $\alpha$-Mg matrix of dendritic morphology with intermetallic compounds $Mg_5Ga_2$ located mainly at the interdendritic regions. The fraction and size of the $Mg_5Ga_2$ particles increased with the amount of Ga in the alloy. The grain size decreased as the Ga content was increased. The products formed on the surface of the Mg-Ga alloys after immersion in SBF were MgO, $Mg(OH)_2$, and calcium phosphates. The corrosion rate of the as-cast alloys was dependent on the Ga content. At concentrations lower than 1 wt. % the corrosion rate was similar to that of pure Mg (0.65 mm/year). However, Ga additions higher than 1 wt. % worsened the corrosion resistance. After heat treatment, the corrosion rate of Mg-Ga alloys decreased, and in the case of the alloys with Ga concentrations lower than 1 wt. %, corrosion rate was lower than that of pure Mg. Corrosion of these alloys after heat treatment was uniform.

**Keywords:** biodegradable alloys; magnesium alloys; gallium; corrosion; precipitation heat treatment

## 1. Introduction

In recent years, Mg alloys as degradable implant materials have become more interesting and are intensively investigated to be implemented as osteosynthesis materials [1–4] due to their mechanical properties, close to those of natural bone, and their biocompatibility [1,2]. It has been demonstrated that Mg alloys implants increase bone mass and mineral apposition rate around the implant [5]. However, rapid corrosion is an intrinsic response of Mg alloys to chloride containing solutions including the human body fluids and blood plasma [6]. This characteristic gives to Mg alloys precisely their biodegradability ability, although Mg-based biomedical implants may lose the necessary mechanical integrity before the tissue has healed completely. This low corrosion resistance produces a rapid formation and accumulation of corrosion products in the surrounding environment [7] such as hydrogen ($H_2$) that causes the formation of bubbles [8]. Hydrogen bubbles formed during the magnesium corrosion process represent a health risk, and localized hydrogen accumulation may cause local alkalization increasing pH in the vicinity of the implant, affecting the physiological processes dependent on pH [8]. In order to sort out the problem of rapid degradation of Mg in the human body and thus be able to use it as a biodegradable material, metallurgical techniques related to their manufacturing processes, such as alloying with other elements and mechanical and heat treatments, have been used [9]. Alloying with other elements is one of the most effective techniques to improve the

corrosion resistance and the mechanical properties of Mg due to the changes produced by the alloying elements in the structure and phases distribution within the Mg matrix [10]. In addition, in biomedical engineering, the biocompatibility of alloying elements needs to be considered. For example, the AE21, AZ21, AZ31, AZ31B, and AZ91D2 alloys have excellent mechanical properties and an acceptable corrosion resistance [10], and many authors have proposed them as biodegradable materials; however, it has been reported that the Al contained in these alloys may cause adverse reactions and toxicity in the body [11]. Other elements, such as Ca [12] and Zr [13], have shown appropriate biocompatibility in vivo and in vitro; however, at high contents of these elements, the corrosion resistance of the alloys decreases [13,14]. In this sense, gallium is an alloying element that is proposed as a candidate to improve the corrosion resistance of magnesium [15] since it exhibits a high hydrogen overpotential and appropriate electrochemical activity [16]; therefore, it can inhibit the cathodic reaction. In addition, Ga is an element that presents an acceptable cellular toxicity profile [17], and it may contribute to bone recovery since it has shown therapeutic activity in metabolic bone disease, hypercalcemia, and cancer [18]. Gallium has also shown, as a compound, efficacy in the osteoporosis treatment; Li et al. [19] demonstrated that gallium nitrate (GaN) counteracts the bone loss in an experimental model of established osteoporosis.

Kubásek et al. [20] analyzed the incorporation of Ga into Mg (1, 4 and 7 Ga, wt. %) preparing as-cast Mg-Ga binary alloys for biomedical purposes. They evaluated both in vitro corrosion (0.9 wt. % NaCl aqueous solution) and cytotoxicity. The results revealed that Ga in concentrations lower than 1wt. % reduced the corrosion rate of Mg, but at higher Ga concentrations, this effect was reversed due to the galvanic effect produced by the second phases with the magnesium matrix. The cytotoxicity tests performed on human osteosarcoma cells showed that there was no alteration of the basic cellular functions.

Mohedano et al. [21] evaluated the corrosion behavior of binary as-cast Mg-Ga alloys (Ga additions from 1 to 4 wt. %) in a 0.5 wt. % NaCl aqueous solution. They found that as the content of the alloying element increased, the corrosion rate was increased, which became more evident for longer immersion times and concentrations of Ga higher than 2 wt. %.

The amount and distribution of the second phases precipitated in the Mg matrix by the effect of the alloying elements can be modified through post-processing techniques such as heat treatment and, therefore, the magnesium corrosion rate can be modified [22–24].

Liu et al. [25] studied a Mg-5.53Ga (wt. %) alloy to evaluate the effect of heat treatment on the mechanical properties of the alloy. The alloy was heat-treated by solution (375 °C, 12 h), quenching and aging (225 °C, 0.5 or 128 h). The mechanical properties of the samples aged for 128 h were similar or slightly lower than those of AZ91E-F HPDC and Mg(0–4)Ca alloys and higher than those of the AZ91 + 2Ca10 alloy.

During the corrosion of Mg and Mg alloys implants, four components are formed: (i) a corroded surface on the implant, (ii) dissolved Mg and other dissolved alloying elements, (iii) a large amount of $OH^-$ and (iv) hydrogen (gas) [26]. The dissolved Mg ends up in two places: the solution and the surface layer. The constituents in this surface layer after corrosion are usually MgO and/or $Mg(OH)_2$ in addition to insoluble phosphates and carbonates [26].

The formation of hydroxyapatite (HA) on these substrates is favored by the dissolution of Mg from the substrate and the pH increase, however, it has been reported that magnesium ions retard or inhibit the HA crystallization and other calcium phosphates under different conditions [27]. Therefore, the interaction of the reaction layer on the substrate of the Mg-Ga alloys with the surrounding medium and its influence on the corrosion rate of the alloy need to be investigated.

Considering the results stated above, in the present work, the effect of the following parameters on the corrosion rate and microstructure of Mg-Ga binary alloys were studied: (a) Ga content, ranging from 0.0 to 1.5 wt. %, (b) effect of the heat treatment (solution–quenching–aging), and (c) effect of the corrosive medium (SBF).

## 2. Materials and Methods

Four binary Mg-Ga alloys were prepared using high purity metals Mg (Stanford Advanced Materials, Lake Forest, CA, USA, 99.99 wt. %) and Ga (Sigma Aldrich, St. Louis, MO, USA, 99.99 wt. %) under controlled atmosphere (Ar-1%SF$_6$) using an electric resistance furnace equipped with a graphite crucible. Either pure Mg or each alloy were melted and then kept at 750 °C for 15 min under stirring for homogenization of the melt. The pure Mg and the alloys solidified under air forced cooling inside the graphite crucible. The chemical composition of the alloys (Table 1) was determined using inductively coupled plasma atomic emission spectroscopy (ICP-OES Perkin Elmer, Boston, MA, USA, model Optima 8300).

**Table 1.** Chemical composition (wt. %) of pure Mg and Mg-Ga alloys (as-cast).

| Nominal Composition | Mg | Ga | Fe | Ni | Cu |
|---|---|---|---|---|---|
| pure Mg | 99.99 | 0 | ≤0.0002 | ≤0.0006 | ≤0.0002 |
| Mg-0.375Ga | 99.57 | 0.37 | – | – | – |
| Mg-0.750Ga | 99.12 | 0.73 | – | – | – |
| Mg-1.125Ga | 98.80 | 1.18 | – | – | – |
| Mg-1.5Ga | 98.56 | 1.43 | – | – | – |

Test specimens of pure Mg and each alloy (10 mm in width, 10 mm in length, and 3 mm in thickness) were machined.

A set of alloy specimens were heat treated. The solution heat treatment (T4) was performed at 350 °C for 12 h under an Ar atmosphere using a muffle furnace (Nabertherm, Lilienthal, Germany, model NBTL40/11/B180). The treated alloys were quenched in water at 25 °C and subsequently were artificially aged (T6) at 225 °C for 16 h under an Ar atmosphere using the muffle furnace and then cooled inside the furnace.

To observe the microstructure, specimens were prepared metallographically. Initially, they were ground (SiC paper, from 600 to 1200 grit size), then polished (diamond paste, 1 and 3 μm) and finally etched. The etching was carried out using acetic-glycol and acetic-picral reagents. The surface of the samples was analyzed by optical microscopy (Olympus Vanox, Center Valley, PA, USA AHMT3) and scanning electron microscopy (SEM, Philips, Houston, TX, USA. XL30 ESEM) with energy dispersive X-ray spectroscopy (EDS) for matrix and second phases characterization.

*Immersion Tests*

The immersion tests were performed using SBF as a medium, which was prepared according to the method proposed by Kokubo [28]. Specimens both as-cast and heat treated were ground (SiC paper, from 320 to 1200 grit size) and then cleaned with acetone in an ultrasonic bath for 15 min.

The measuring of the corrosion rate was performed by the weight loss method, following the stated in the G31-72, G1, and G31 ASTM standards [29–31]. Clean and dry specimens were weighted using an analytical balance (Ohaus, accuracy of 0.0001 g), and then each one of them was immersed in 30 ml of SBF contained in a plastic flask. The flasks were kept at 37 °C (±0.5) in an incubator (Fisher Scientific, Waltham, MA, USA, model 637D) for immersion times of 7, 14, 21, and 28 days. After each immersion period, the specimens were removed from the SBF and immersed in a solution (200 g/L of CrO$_3$, 10 g/L of AgNO$_3$ and 20 g/L of Ba(NO$_3$)$_2$) [30] for 15 min to remove the corrosion products. Specimens were cleaned with alcohol using an ultrasonic bath for 15 min and dried. Specimens were weighted again in order to evaluate the weight loss using Equation (1) [30]:

$$\text{corrosion rate} = \frac{8.76 \times 10^4 \, W}{ATD} \tag{1}$$

where:

$T$ = exposure time (h)
$A$ = specimen area (cm$^2$)
$W$ = mass loss (g)
$D$ = density (g/cm$^3$)

Specimens density was measured using the Archimedes principle. The topography of the corroded specimens after the different immersion periods was observed by SEM and the phases were analyzed by EDS.

In order to identify the corrosion products and other formed compounds on the samples after the immersion test, a surface analysis by X-ray thin film diffraction within the range of 10° and 80° in the 2θ position was performed (Bruker, Billerica, MA, USA, D8 Advance). Fourier transform infrared spectroscopy (FTIR, Fisher Scientific, Waltham, MA, USA, Nicolet iS5) was used as a complementary technique.

The chemical composition of the remaining SBFs was measured by inductively coupled plasma atomic emission spectroscopy (ICP-OES, Perkin Elmer, Boston, MA, USA, model Optima8300). Additionally, pH of SBFs was evaluated (Fisher Scientific, Waltham, MA, USA, Orion Star A211).

## 3. Results and Discussion

The microstructure of the as-cast Mg-Ga alloys (Figure 1) consists of a primary α-Mg matrix (light gray zone) of dendritic morphology and a second phase constituted of precipitates mainly distributed in the interdendritic regions and grain limits (dark zones). According to the binary Mg-Ga phase diagram, these precipitates correspond to the Mg$_5$Ga$_2$ intermetallic. The precipitates vary in size and proportion depending on the amount of gallium in the different alloys; the Mg-0.375Ga alloy (Figure 1a) contains the lower amount of precipitates, while the Mg-1.5Ga alloy (Figure 1d) contains the higher amount. In this last alloy, precipitates form almost continuous lines; this behavior is attributed to the fact that the solid solubility of gallium into magnesium at room temperature is very limited.

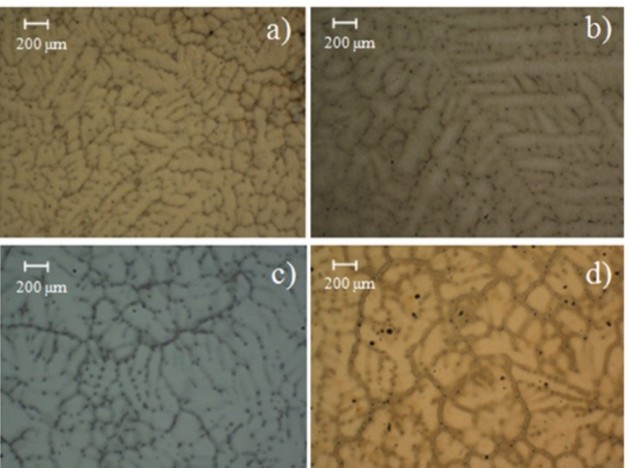

**Figure 1.** Optical micrographs showing the microstructure of (**a**) Mg-0.375Ga, (**b**) Mg-0.75Ga, (**c**) Mg-1.125Ga, and (**d**) Mg-1.5Ga as-cast alloys.

The morphology of precipitates in the analyzed alloys (Figure 2) is also dependent on the Ga content. Alloys with less than 1 wt. % of Ga show precipitates with predominantly semi-spherical morphology (Figure 2a), while alloys with higher Ga content show precipitates of elongated strip morphologies (Figure 2b). Figure 2 shows also the semiquantitative analysis (EDS, at. %) of the precipitates and the matrix. Analysis of $X_A$ and $X_C$ zones confirms that precipitates correspond to the Mg$_5$Ga$_2$ intermetallic. Analysis of $X_C$ and $X_F$ zones corresponds to the Mg matrix, and analysis of $X_B$ and $X_D$ zones corresponds to regions between matrix and precipitates.

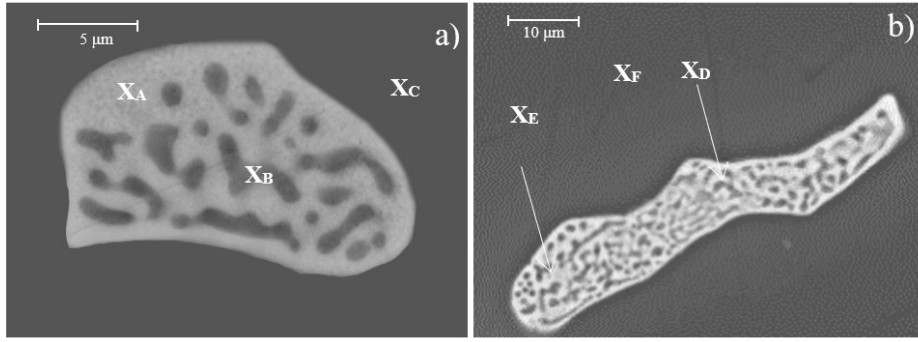

| Zone | Mg, at. % | Ga, at. % |
|------|-----------|-----------|
| $X_A$ | 71.82 | 28.18 |
| $X_B$ | 89.31 | 10.69 |
| $X_C$ | 99.46 | 0.54 |
| $X_D$ | 87.58 | 12.42 |
| $X_E$ | 72.09 | 27.91 |
| $X_F$ | 98.68 | 1.32 |

**Figure 2.** SEM images of precipitates in the as-cast Mg-Ga alloys, (**a**) Mg-0.375Ga alloy, (**b**) Mg-1.5Ga alloy, and corresponding EDS analysis.

Table 2 shows the average grain size of pure Mg and as-cast alloys. As observed, grain size depends on the Ga amount in the alloy; as the amount of Ga is increased, the grain size decreases. This fact indicates that Ga is an effective grain refiner [21].

**Table 2.** Average grain size of pure Mg and as-cast Mg-Ga alloys.

| Alloy | Average Grain Size (μm) |
|-------|-------------------------|
| pure Mg | 530 |
| Mg-0.375Ga | 486 |
| Mg-0.750Ga | 360 |
| Mg-1.125Ga | 216 |
| Mg-1.5Ga | 192 |

Figure 3 shows SEM images of the Mg-1.125Ga and Mg-1.5Ga alloys before and after heat treatment. The as-cast alloys microstructure (Figure 3a,c) consists of two zones: A dark zone corresponding to the α-Mg matrix and bright zones corresponding to the intermetallic precipitates. After heat treatment, alloy microstructures were modified obtaining finer and more homogeneously distributed particles (Figure 3b,d). During the solution heat treatment, intermetallics were dissolved into the matrix through a diffusion mechanism activated by temperature, reaching a more homogeneous Ga distribution. When the alloy is quenched, Ga is maintained in solution (non-equilibrium state), and then, during artificial aging, controlled precipitation of the intermetallics is promoted.

The efficiency of the solution heat treatment on the dissolution and redistribution of the alloying element is a function of the amount of gallium added, and it is observed after the aging treatment. For alloys with Ga content lower than 1 wt. %, the precipitated intermetallics are so small and are so well distributed that they are not be observed (Figure 3b). As the Ga content increases above 1 wt. %, precipitates size starts to increase and their distribution starts to be heterogeneous (Figure 3d).

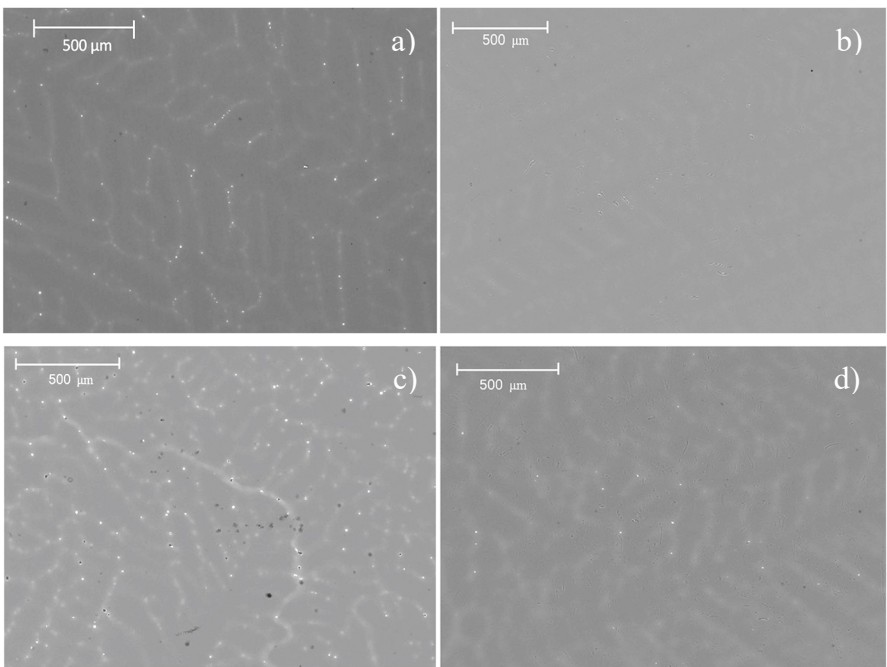

**Figure 3.** BSE-mode SEM images of the microstructure of the alloys: (**a**) as-cast Mg-1.125Ga, (**b**) heat treated Mg-1.125Ga, (**c**) as-cast Mg-1.5Ga, (**d**) heat treated Mg-1.5Ga.

Corrosion rate and pH of corresponding SBF for pure Mg and Mg-0.375Ga and Mg-1.5Ga alloys as a function of immersion time are presented in Figure 4. As observed, corrosion rate decreased, while pH increased as the immersion period was increased. This behavior was similar for all the studied alloys. At the initial stage of immersion, pH of SBF was 7.4 and under this condition Mg is rapidly corroded. As Mg is dissolved, pH is increased due to the released $OH^-$ ions until reaching a value close to 10. This pH value is close to 10.2 where the $Mg(OH)_2$ compound becomes stable. This layer acts as a barrier between the magnesium substrate and the corrosive medium, decreasing corrosion rate [32].

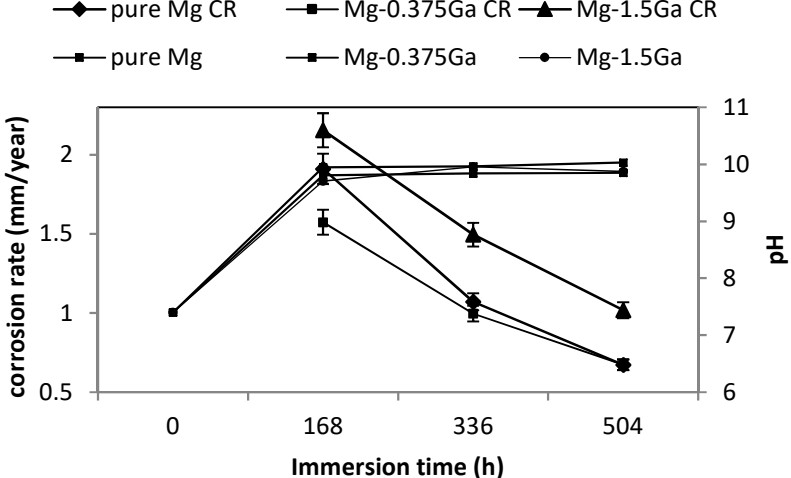

**Figure 4.** Corrosion rate and pH of the corresponding SBF for pure magnesium, Mg-0.375Ga, and Mg-1.5Ga alloys as a function of immersion time.

Figure 5 shows the XRD patterns of pure Mg and Mg-Ga alloys after 168 and 672 h of immersion in SBF. At early stage (168 h of immersion, Figure 5a), the predominant phases are α-Mg and MgO. As immersion time increases, peaks corresponding to $Mg(OH)_2$ were identified. This last fact was

expected due to the increase of pH in the solution. The presence of other phases such as Ca,P-rich compounds was not detected by this technique.

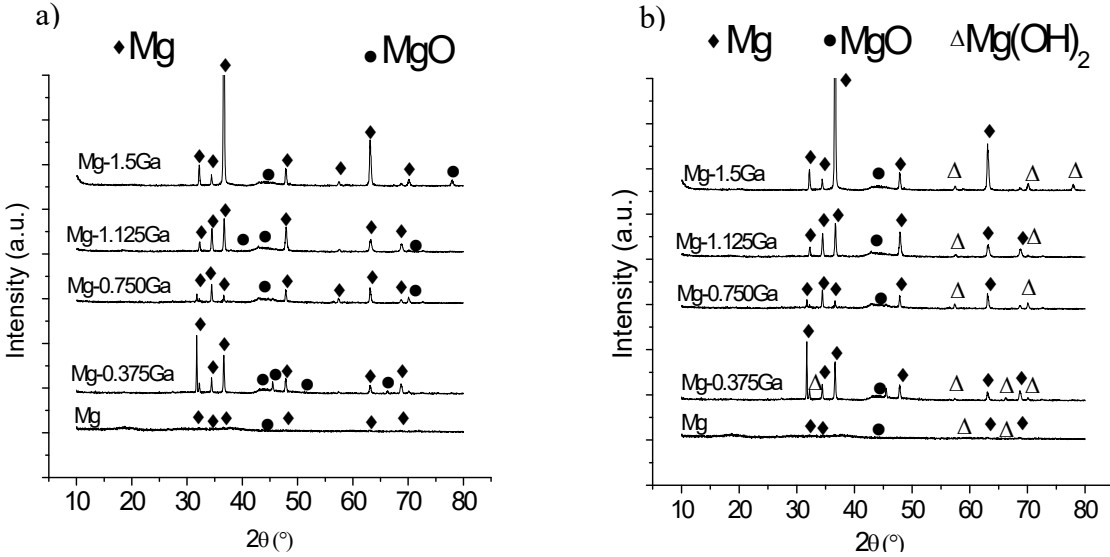

**Figure 5.** XRD patterns of pure Mg and Mg-Ga alloys after (**a**) 168 and (**b**) 672 h of immersion in SBF.

In addition to the MgO and Mg(OH)$_2$ phases detected by XRD on the samples, particles and agglomerates were also observed using SEM. Figure 6 shows SEM images of the Mg-0.375Ga alloy after 168 h of immersion in SBF. Particles of spherical morphology (approximately 1 μm in diameter) and agglomerates of these (5–800 μm) were observed. This morphology resembles that of the apatite formed on bioactive systems [33,34].

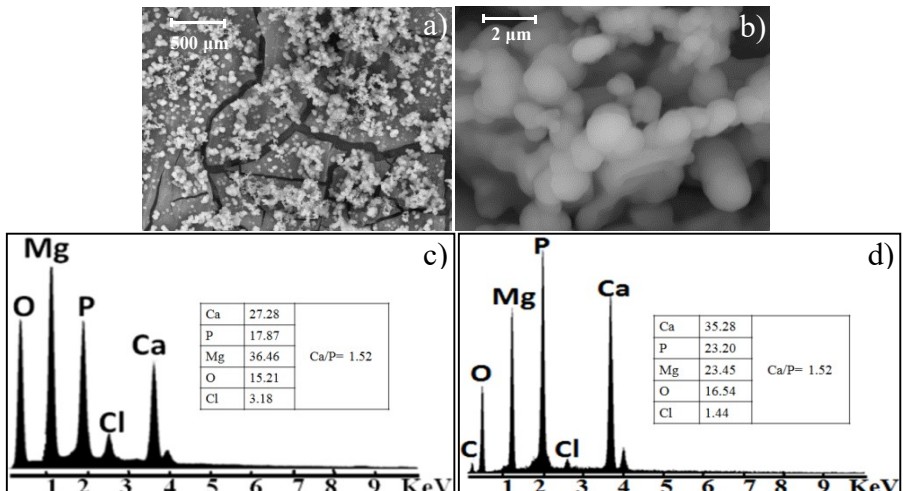

**Figure 6.** SEM images and corresponding EDS analysis of the Mg-0.375Ga alloy after 168 h of immersion in SBF: (**a**) 50× and (**b**) 5000×.

According to the EDS spectra (Figure 6b,c), particles are mainly constituted by Mg, Ca, and P with a Ca/P atomic ratio of 1.52. This atomic ratio is consistent with that of some calcium phosphates [33] that can be formed on magnesium substrates and its alloys such as tricalcium phosphates, dicalcium phosphate, amorphous calcium phosphates, and hydroxyapatite [35,36]. These Ca, P-rich compounds improve the biocompatibility of metallic implants and increase bone growth at the site of implantation [37]. A bioactive material has the ability to bond to bone through an

apatite layer. It has been shown that this apatite layer can also be reproduced on bioactive materials by immersing them in SBF. The nucleation of the Ca,P-rich compounds occurs at the surface of the substrates (Mg or Mg-alloys), and then these nuclei grow, both events at the expense of the Ca and P ions of the SBF, until forming the layer observed in Figure 6b. According to the EDS results, the calcium phosphates formed on the substrates have a high substitution of Ca by Mg.

It has been reported [38] that the characteristic reflections of the $PO_4^{3-}$ vibrations are located at 460, 560–600, 960, and 1020–1120 cm$^{-1}$. Figure 7 shows FTIR spectra of pure Mg and the Mg-Ga alloys after 168 h of immersion in SBF and the FTIR spectrum of the synthetic hydroxyapatite (HA). The characteristic reflections of these vibrations (at 560, 600, 630, and 1020 cm$^{-1}$) were observed on both HA and the Mg-Ga alloys. These results agree with the EDS analyses (Figure 6) and confirm the nature of the formed compounds on the samples after immersion in SBF.

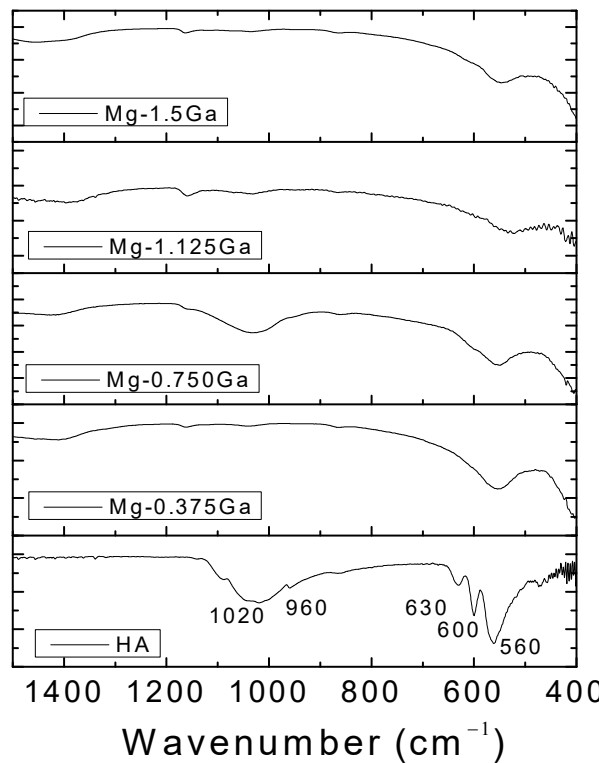

**Figure 7.** FTIR spectra of pure magnesium and Mg-Ga alloys after 168 h of immersion in SBF and FTIR spectrum of synthetic hydroxyapatite (HA) from 1400 to 400 cm$^{-1}$.

Figure 8 shows the ionic concentration of Ca, P, and Mg in the remaining SBFs as a function of immersion time for the Mg-0.375Ga alloy. As observed, Ca and P concentrations decreased as the immersion time was increased, which is attributed to the formation of calcium phosphates on the metallic samples. As expected, the concentration of Mg increased as the immersion time was increased due to the magnesium dissolution. A similar behavior was observed for all the remaining SBFs.

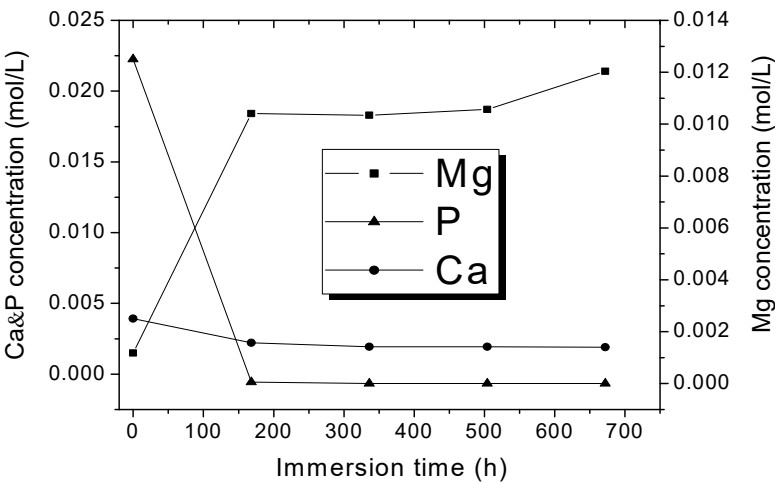

**Figure 8.** Ionic concentration of remaining SBFs.

The corrosion rate of pure magnesium and those of the different as-cast alloys after immersion in SBF for 504 h are shown in Figure 9. The value for pure magnesium was 0.65 mm/year and the corrosion rates for Mg-0.375Ga and Mg-0.75Ga alloys were similar. However, for Mg-1.125Ga and Mg-1.5Ga alloys, the corrosion rate increased significantly to values close to 1 mm/year. For high levels of Ga, corrosion rate followed a tendency to increase as the Ga content was increased.

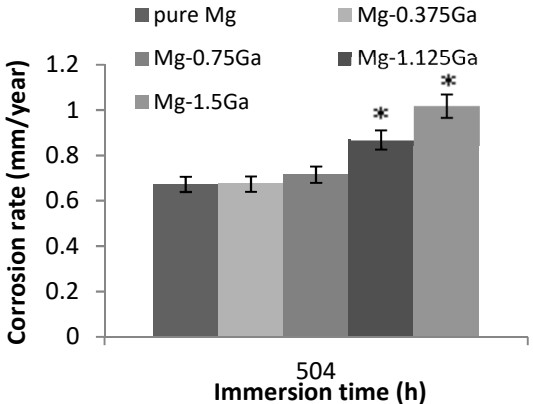

**Figure 9.** Corrosion rate obtained by weight loss for pure magnesium and the different as-cast alloys after immersion in SBF for 504 h (*Statistically higher than 0.65 mm/year, $p > 0.5$).

The metallic materials topography (once the corrosion products have been removed) is shown in Figure 10. Pure magnesium showed uniform corrosion type with low depth elongated depressions (Figure 10a), while Mg alloys presented pitting corrosion (Figure 10b,c). As it can be observed, cavities in the Mg-0.375Ga alloy (Figure 10b) are smaller and less deep than those observed in the Mg-1.5Ga alloy. The increase in the corrosion damage observed on the as-cast alloys with the increase in Ga concentration is consistent with the previously determined corrosion rates, where an increase in the corrosion rate was observed as a function of the increase in the amount of Ga.

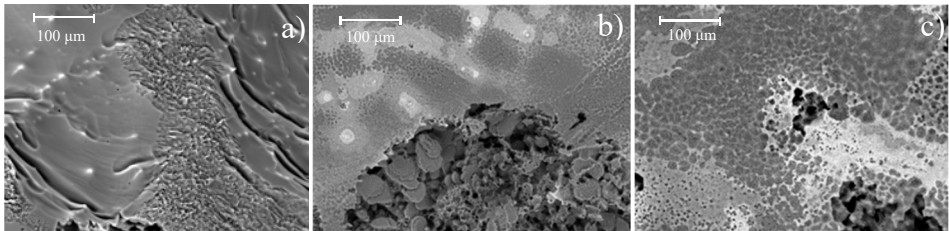

**Figure 10.** SEM images of the topography of (**a**) pure magnesium and (**b**) Mg-0.375Ga and (**c**) Mg-1.5Ga as-cast alloys after 168 h of immersion in SBF.

Figure 10 shows SEM images of the Mg-0.375Ga alloy topography after corrosion testing. At lower magnifications (Figure 11a), some shallow and isolated pitting was observed. A close-up of one cavity adjacent to a precipitate is shown in Figure 11b. The bright white dot in this image corresponds to $Mg_5Ga_2$ (identified by EDS analysis, Figure 11c).

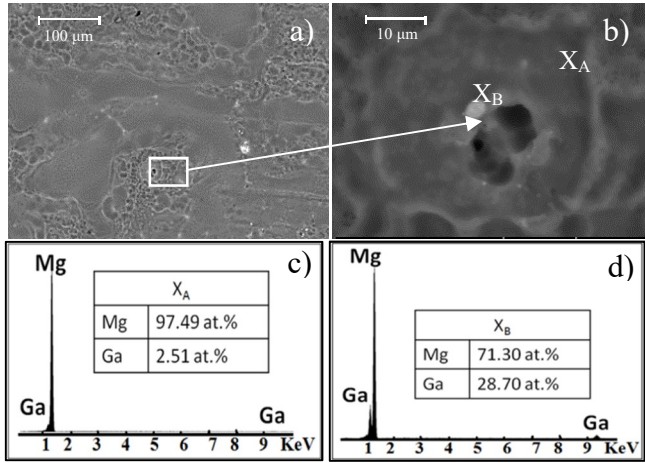

**Figure 11.** SEM images of the topography of (**a**) as-cast Mg-0.375Ga after 504 h of immersion in SBF, (**b**) a detail of pitting adjacent to $Mg_5Ga_2$ precipitate, (**c**) EDS analysis of $X_A$ zone, (**d**) EDS analysis of $X_B$ zone.

In the EDS analyses (Figure 11c,d), a difference in composition between the matrix and the precipitates can be distinguished—the precipitates contain a higher amount of Ga in comparison to the matrix. This increase in Ga in the second phase indicates that microgalvanic corrosion occurs between $\alpha$-Mg and precipitates. Magnesium is located at the most active end of the galvanic series and it has lower potential than Ga. Thus, Mg behaves as an anode and it corrodes itself in an accelerated way. This galvanic corrosion generates the selective dissolution of the matrix surrounding the precipitates showing a pitting morphology [32].

Figure 12a shows the corroded surface of the Mg-1.5Ga alloy at low magnifications. As it can be observed, cavities are larger and deeper than those in the Mg-0.375Ga alloy (Figure 10a). Figure 12b shows a close-up of the corroded surface where the dissolution of the matrix in the area surrounding the precipitate ($X_B$) is observed. Chemical semiquantitative analysis of the precipitate ($X_B$) and the surrounding area ($X_C$) are presented in Figure 12d,c, respectively. It was observed that cavities increased in size and depth as a function of Ga content due to the amount and size of $Mg_5Ga_2$ precipitates.

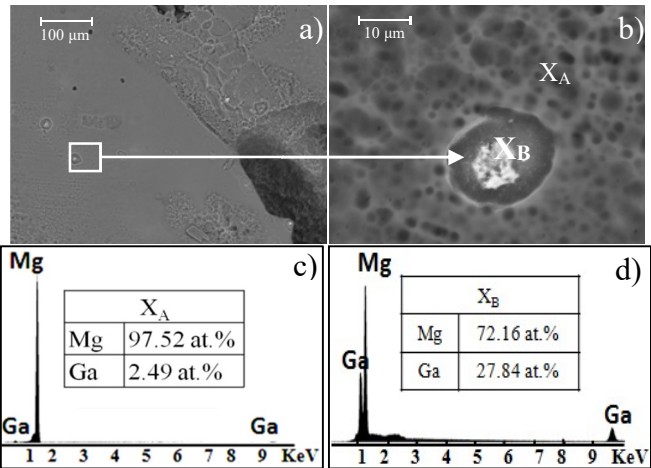

**Figure 12.** SEM images of the topography of (**a**) as-cast Mg-1.5Ga after 504 h of immersion in SBF, (**b**) close-up of pitting adjacent to $Mg_5Ga_2$ precipitate, (**c**) EDS analysis $X_A$ zone, (**d**) EDS analysis of $X_B$ zone.

The Mg-0.375Ga and Mg-0.75Ga as-cast alloys exhibited a corrosion rate close to that of pure magnesium and were selected for heat treating. The measured corrosion rate for these heat-treated alloys, as well as that of pure magnesium after 504 h of immersion in SBF, is shown in Figure 13. As observed, corrosion rate decreased after heat treatment (8.6% and 16.1% decrease in corrosion rate, respectively).

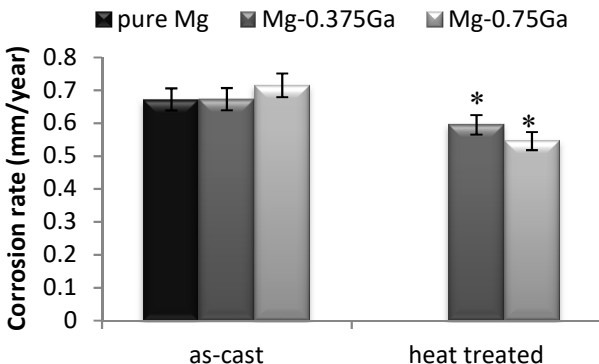

**Figure 13.** Corrosion rates before and after heat treatment for pure magnesium and Mg-Ga alloys after 504 h of immersion in SBF (*Statistically lower than 0.65 mm/year, $p > 0.5$).

This improvement is related to the precipitates redistribution ($Mg_5Ga_2$) as a result of the heat treatment; precipitates are finer and homogeneously distributed, inducing more homogeneous corrosion and decreasing corrosion rate.

Figure 14a shows the topography of the heat-treated Mg-0.375Ga alloy after 168 h of immersion in SBF. Uniform corrosion with the absence of pitting can be observed. This SEM image corroborates that both decrease in size and redistribution of the $Mg_5Ga_2$ intermetallic particles were able to decrease the galvanic corrosion. Figure 14b shows the topography of the heat-treated Mg-0.75Ga alloy after 168 h of immersion in SBF. As it can be seen, corrosion started at the grain boundaries and the interdendritic regions. In both areas, small cavities are observed, smaller than those observed on the corresponding as-cast alloy. This corrosion behavior is related to the cathodic effect generated by the difference in concentration between precipitates and matrix in the as-cast alloys [39]. On the other hand, the precipitates in this alloy after heat treatment were refined and redistributed, so the resulting cavities were smaller.

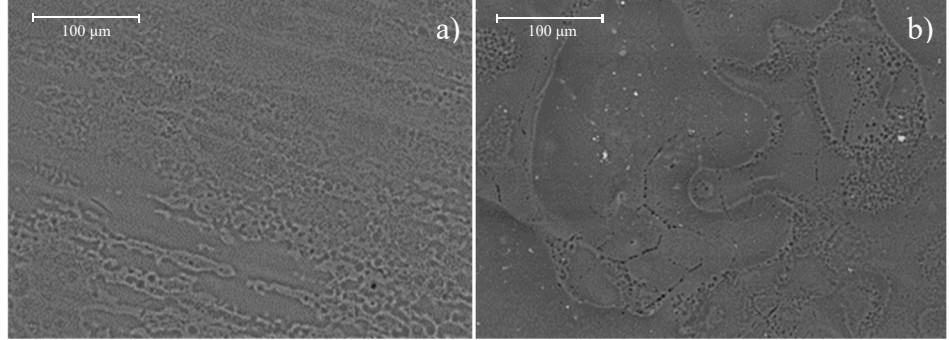

**Figure 14.** SEM images of the topography of heat-treated (**a**) Mg-0.375Ga, (**b**) Mg-0.750Ga after immersion in SBF for168 h.

## 4. Conclusions

Different Mg-Ga alloys were obtained. The amount and size of the $Mg_5Ga_2$ precipitates observed in the as-cast alloys were directly related to the amount of Ga in the alloy. Gallium also acted as an effective grain refiner. After heat treatment, the alloy microstructures were modified obtaining finer and more homogeneously distributed precipitates.

The corrosion rate of the as-cast alloys with Ga content lower than 1 wt. % was similar to that of pure Mg (0.65 mm/year). However, for those with Ga content higher than 1 wt. %, corrosion rate was higher than that of pure Mg (close to 1 mm/year). In all the cases, the corrosion mechanism observed was that of the galvanic type (pitting).

For the heat-treated alloys with Ga content lower than 1 wt. %, the corrosion rate was lower than that of pure Mg (between 8.6% and 16.1% lower). In this case, corrosion was more uniform. Both facts, the decrease in corrosion rate and the change in the corrosion mechanism, are beneficial for degradable alloys for biomedical applications.

After immersion of the metallic substrates in SBF, corrosion products (MgO and $Mg(OH)_2$) and Ca,P-rich compounds were detected. The presence of these calcium phosphates indicates the bioactivity of the alloys developed in this work.

**Author Contributions:** J.C.E. and A.A.H. conceived and designed the experiments; A.A.H. performed the experiments; J.C.E., A.A.H., D.A.C.; and J.M.A. wrote the paper; and all the authors analyzed the data and provided comments and corrections to the manuscript.

**Funding:** This research was financed by CONACYT, México.

**Conflicts of Interest:** The authors declare no conflicts of interest.

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
