# Peer review of "Effect of Gallium Content and Heat Treatment on the Microstructure and Corrosion Rate of Magnesium Binary Alloys"

_metals, doi:10.3390/met9090990_

Round 1
Reviewer 1 Report
The manuscript is well constructed and, although the methodology is quite simple, the results are well presented and the discussion is scientifically sound. The manuscript allows for a quick reading and easy comprehension of the subject. Even though the subject researched is not particulalry new there are still some interesting points. I would only recomend that the authors correct the formatting of their images, since in many occasions the numering (a), (b), etc is not properly positioned.
Author Response
Point 1: I would only recommend that the authors correct the formatting of their images, since in many occasions the numbering (a), (b), etc is not properly positioned
Comment: Agreed. The images format has been corrected in the revised version as suggested.
Reviewer 2 Report
Article is very interesting and written very well. The methodology and results was very well presented.
Comments and Suggestions for Authors - the proposed amendments:
The size and contrast of some photos could be improved.
A slight correction of the title of the article is suggested. The authors should use the wording "corrosion rate".
Designations of drawings (a, b, c, d) - illegible, use drawings in tables.
Fig. 2. - black font in the pictures - illegible.
Fig. 4. - lack of standard deviation.
Fig. 5. - incorrect marking of drawings "a" and "b" (similarly as before).
Fig. 6. - incorrect marking of drawings "c" and "d".
Fig. 10. - designations of drawings (a, b, c) - illegible, use drawings in tables.
Fig. 11., Fig. 12. - designations of drawings (a, b, c, d) - illegible, use drawings in tables.
Fig. 14. - the lack of signs/marks "a" and "b".
Author Response
Point 1: The size and contrast of some photos could be improved.

Comment 1: Agreed and done. The resolution of all the images was modified; the size of images 1-3 and 14 was also modified.
Point 2: A slight correction of the title of the article is suggested. The authors should use the wording "corrosion rate".
Comment 2: Agreed and done. The title was changed to: Effect of gallium content and heat treatment on the microstructure and corrosion rate of magnesium binary alloys
Point 3: Designations of drawings (a, b, c, d) - illegible, use drawings in table
Comment 3: Agreed and done